# The Flashlight-Sign: A Novel B-Flow Based Ultrasound Finding for Detection of Intraluminal, Wall-Adherent, Floating Structures of the Abdominal Aorta and Peripheral Arteries

**DOI:** 10.3390/diagnostics12071708

**Published:** 2022-07-13

**Authors:** Christian Lottspeich, Daniel Puhr-Westerheide, Jan Stana, Ulrich Hoffmann, Michael Czihal

**Affiliations:** 1Division of Vascular Medicine, Medical Clinic and Policlinic IV, Hospital of the Ludwig-Maximilians-University, 80336 Munich, Germany; christian.lottspeich@med.uni-muenchen.de (C.L.); ulrich.hoffmann@med.uni-muenchen.de (U.H.); 2Interdisciplinary Ultrasound Department, Medical Clinic and Policlinic IV, Hospital of the Ludwig-Maximilians-University, 80336 Munich, Germany; 3Clinic and Policlinic for Radiology, Hospital of the Ludwig-Maximilians-University, 80336 Munich, Germany; daniel.puhr-westerheide@med.uni-muenchen.de; 4Department of Vascular Surgery, Hospital of the Ludwig-Maximilians-University, 80336 Munich, Germany; jan.stana@med.uni-muenchen.de

**Keywords:** abdominal aorta, arterial embolism, B-Flow sonography, contrast enhanced ultrasound, endovascular aortic repair, flashlight sign, peripheral arteries, wall adherent floating structures

## Abstract

This study aimed to evaluate the potential diagnostic value of a novel, sonographic, B-Flow (BFl)-based sign (“flashlight sign”, FLS) for the detection of wall-adherent, floating arterial structures (WAFAS). The FLS, characterized by a fast moving, very bright, intraluminal signal, was detected in 28 patients with WAFAS. We divided this cohort into three subgroups according to the affected vascular segments: (1) peripheral arteries (*n* = 10); (2) native abdominal aorta (*n* = 8); and (3) abdominal aorta after endovascular aortic repair (EVAR; *n* = 10). Clinical characteristics were analyzed and BFl-findings were compared with contrast-enhanced ultrasound (CEUS) and computed tomography angiography (CTA). Seven patients (25%) suffered from arterial embolism downstream to the FLS (EVAR, *n* = 4; native abdominal aorta, *n* = 1; peripheral arteries, *n* = 2). WAFAS of the abdominal aorta (native or after EVAR), as indicated by the FLS, were visible by CEUS and CTA in 60% and 93.3%, respectively. Based on the largest cohort (to this point) of patients with WAFAS, we propose a clinically useful, BFl-based sonographic sign for the detection of these underrated arterial pathologies in the abdominal aorta and the peripheral arteries.

## 1. Introduction

Mural thrombus of the aorta is usually diagnosed by cross-sectional imaging in patients suffering from peripheral arterial embolism or as an incidental finding in asymptomatic patients [1]. In a large autopsy series, mural thrombus of the aorta was found in 0.45% of cases [2]. In abdominal aortic aneurysms, mural thrombus can be observed much more frequently (>90% in large aneurysms with indication for repair [3]. After endovascular aortic repair (EVAR), wall-adherent thrombus is also a common finding (up to 20% of patients), and is mainly detected within the first year after implantation [4]. However, the absolute risk of peripheral embolization related to mural thrombus within an abdominal aortic aneurysm or an EVAR-prosthesis seems to be rather low [4].

Mobile or floating thrombus adherent to the arterial vessel wall (here referred to as wall adherent floating arterial structures, WAFAS) has been reported less frequently, so far predominantly described in the descending thoracic aorta and the carotid arteries [5,6]. The actual risk of peripheral embolization of aortic WAFAS is poorly defined, but may be substantially higher than that of non-mobile mural thrombus [7]. In about 0.2–1.3% of patients suffering from ischemic stroke, an ipsilateral carotid free-floating thrombus can be found [5,8].

Diagnostic imaging of WAFAS is challenging. Colour duplex sonography is the primary diagnostic tool for evaluation of the abdominal aorta and the peripheral arteries and offers high spatial resolution as well as dynamic information on the blood flow. Contrast-enhanced ultrasound (CEUS) with second generation contrast agents provides additional information with regard to the detection of slow blood flow, vessel wall perfusion and intraluminal filling defects. B-Flow (BFl) is a patented technology developed by General Electric Medical Systems, Milwaukee, WI, USA (GE) [9]. Unlike colour duplex sonography, BFl is not based on the Doppler principle. By using tissue-blood equalization technology, BFl detects moving objects by displaying the difference between two pictures of the object taken at different time points. Consequently, BFl can detect motion within anatomical structures with a spatial resolution similar to that of gray scale imaging. The magnitude of difference (as determined by the time interval between pictures and the speed of the object) is correlated to the brightness of the signal, making this method potentially useful for the detection of WAFAS [9,10].

Against this background, the present study aimed to determine the potential diagnostic yield of BFl in the detection of WAFAS in various anatomical locations and to compare BFl with CEUS and computed tomography angiography (CTA).

## 2. Patients and Methods

Consecutive patients with a sonographic diagnosis of WAFAS in any arterial segment diagnosed between January 2016 and December 2021 were retrospectively identified. Patient’s records were screened for clinical characteristics, sonographic findings (including BFl and CEUS) and CTA performed for the diagnostic workup of these vascular pathologies.

Two experienced vascular sonographers (C.L. and M.C.) consented the WAFAS-diagnosis based on independent review of recorded movies of ultrasound examinations of the abdominal aorta, the supraaortic arteries and the extremity arteries, including BFl. A characteristic BFl-finding was discovered and its potential diagnostic role for detection of WAFAS was analyzed in comparison to established imaging modalities.

All ultrasound examinations were conducted with a GE LOGIQ E9 ultrasound machine using a 4–10 MHz linear transducer (for examination of the supraaortic and extremity arteries) and/or a 2–6 MHz convex probe (for examination of the abdominal aorta and the iliac arteries). BFl was performed in all patients for assessment of the floating intraluminal structures in the longitudinal and transversal plane. For BFl, flash reduction was set on zero by default. The pulse repetition interval was 10 (linear probe) and 18 (convex probe), respectively. On a detection speckle reduction imaging strength scale ranging from 0 to 4, the values were set to 1 (linear probe) and 2 (convex probe), respectively. When CEUS was performed (on the same of the BFL-study), machine settings were adjusted appropriately (low MI imaging, as defined previously [11] and movies of the respective vascular segment were recorded after injection of 1.2 mL sulfur hexafluoride intravenously.

CTA images of the suspicious segments and, if indicated, of the upstream and/or downstream arterial tree, were independently reviewed by an experienced radiologist who was unaware of the clinical and sonographic data (D.P.-W.). Different CT scanners (Siemens SOMATOM Force, Siemens SOMATOM Definition AS+, GE Discovery CT750 HD, GE Optima CT660) were used for image acquisition with single (arterial) phase CTA or multiphasic CTA including an arterial phase. 1–1.5 mL/kg body weight iodinated contrast agent was administered at a flow rate of 3–5 mL/s followed by a 100 mL saline flush at 3–5 mL/s and the arterial phase images were acquired. The arterial phase was used for the assessment of wall adherent thrombus material.

Explorative data analysis was focussed on previous revascularization procedures and symptoms/signs of peripheral arterial embolism downstream to the vessel segments displaying the FLS. The most probable source of embolism in symptomatic patients suffering from arterial embolism was defined based on the results of cardiac diagnostics (Holter electrocardiogram, echocardiography) and sonographic and computed tomography imaging. Depending on the location of the WAFAS in the native abdominal aorta, in the limbs of EVAR prostheses, or in peripheral arteries (supraaortic arteries and extremity arteries) the cohort was divided into three subgroups. These were compared with regard to clinical and imaging findings.

## 3. Results

### 3.1. The Flashlight Sign and Cohort Characteristics

Twenty-eight patients were included in the analysis (85.7% men, mean age 72 ± 8.3 years), all of whom exhibited a BFl-based vascular ultrasound finding characterized by a fast moving, very bright intraluminal signal and indicating the presence of a WAFAS. Given the characteristic appearance, we termed this novel sonographic finding *flashlight sign (FLS)* (Figure 1, Appendix A).

Seven patients in the cohort suffered from thromboembolic arterial events. In all of these patients the WAFAS, as visualized by the FLS in BFl, was deemed to be the most likely source of arterial embolism (Table 1). In the remaining patients, the FLS was detected incidentally (*n* = 7) or during routine surveillance following revascularization procedures (*n* = 14). Characteristics of the subgroups with respect to the anatomic location of the FLS are outlined below and summarized in Table 2.

Seventeen and thirteen out of 28 patients underwent CEUS and CTA, respectively. The mean time interval between the sonographic study (BFl with or without CEUS) and the CTA was 11.3 ± 15.2 days. When comparing BFl with CEUS and CTA, positive predictive values of the FLS for detection of WAFAS were 76.5% and 61.5%. However, there were remarkable differences between subgroups based on the affected arterial segments (native abdominal aorta, abdominal aorta after endovascular repair, peripheral arteries, see Table 2).

### 3.2. Patients with Pathologies of the Native Abdominal Aorta

The FLS was detected in eight male patients in the native infrarenal aorta, three of whom had abdominal aortic aneurysms (Figure 1, Appendix A). One of these patients suffered from unilateral embolic popliteal artery occlusion and in two patients the FLS was detected after preceding endovascular procedures involving catheter passage of the abdominal aorta. In four patients, the FLS was detected incidentally. Another female patient who had undergone the open repair of the infrarenal aorta decades ago presented with thrombotic material in the suprarenal aorta and bilateral renal artery stenoses; in this patient an FLS was detected incidentally in the suprarenal aorta.

CEUS-imaging was performed in only one patient but was negative regarding the presence of a WAFAS. CTA was performed in six patients, proving the presence of wall adherent intraluminal structures surrounded by contrast agent, indicative of WAFAS.

### 3.3. Patients after Endovascular Aortic Repair (EVAR)

The FLS was found in 10 male patients who had undergone EVAR of an infrarenal abdominal aortic aneurysm, four of whom presented with embolic occlusions of the popliteal and/or below the knee arteries. In six patients, the FLS was detected during routine surveillance after EVAR (Figure 2, Appendix A).

The FLS was detected in the right and left EVAR limb in four patients, respectively, and two patients exhibited the FLS in both EVAR limbs. All embolic occlusions were located ipsilaterally to the FLS (three patients with unilateral embolism, one patient with bilateral FLS who developed metachronous bilateral popliteal artery embolism). In another patient who underwent endovascular thrombectomy for right-sided embolic popliteal artery occlusion, the FLS of both EVAR limbs disappeared after four weeks of anticoagulation with low molecular weight heparin. After discontinuation of low molecular weight heparin, the patient suffered from embolic re-occlusion of the right popliteal artery, and a prominent FLS corresponding to a large floating intraluminal thrombus was detected in the right EVAR limb.

Nine patients also underwent CEUS-imaging, which detected intraluminal floating structures in six patients (66%). CTA was performed in eight patients and allowed for the detection of wall-adherent structures within the EVAR-prosthesis in seven cases.

### 3.4. Patients with Pathologies of the Peripheral Arteries

The FLS was detected in peripheral arteries of 10 patients (seven men and three women), including the femoral arteries in six patients, the carotid arteries in two patients, and the brachiocephalic artery and axillary artery in one patient, respectively. In two patients, the FLS-positive lesion was considered the most probable source of symptomatic arterial embolism. One female patient with right-sided retinal artery occlusion had a large arteriosclerotic plaque with WAFAS of the brachiocephalic trunk. Another male patient suffered from left hand ischemia due to an arterial embolism originating from a stenotic arteriosclerotic lesion of the ipsilateral axillary artery containing a WAFAS. It is worthy of note that in six patients the FLS was detected after revascularization procedures (two FLS of the common femoral artery which had been used as an access vessel for endovascular procedures, three FLS of the superficial femoral artery after endovascular recanalization, and one FLS after carotid endarterectomy). Two FLS were found incidentally in patients with severe calcifying arteriosclerosis (origin of the internal carotid artery and of the superficial femoral artery, respectively).

CEUS-imaging and CTA was performed in three patients, respectively. In two CEUS-examinations but in none of the CTA-examinations the WAFAS could be confirmed.

### 3.5. Clinical Impact of the FLS

In six patients the FLS had a direct, additional impact on further patient management, including the initiation of anticoagulation in four patients and the initiation of antiplatelet therapy in two patients. One asymptomatic patient with a FLS corresponding to WAFAS adherent to a mid-grade internal carotid artery stenosis (Figure 3) underwent successful carotid endarterectomy. The above-mentioned patient with recurrent popliteal artery embolism originating from a floating intraluminal thrombus located in the EVAR limbs underwent endovascular re-lining of the prosthesis with covered stent grafts.

### 3.6. Follow Up

Repeated BFl-imaging was performed in 15 patients, with a mean follow-up time of 10 months (range 1 to 35 months). Five patients underwent revascularization of the arterial segment displaying the FLS, resulting in the disappearance of the FLS in all of them. The FLS persisted in five patients (EVAR, *n* = 3; native abdominal aorta, *n* = 2) and disappeared in five patients (EVAR, *n* = 1; peripheral arteries, *n* = 4) treated conservatively.

## 4. Discussion

Based on a novel sonographic sign in BFl, which we termed FLS, we characterized the so far largest cohort of patients with WAFAS in various locations. The FLS may indicate an increased recurrence risk in patients who suffered from symptomatic embolism of the arterial tree downstream to the lesion and thus may have a direct impact on the patient’s treatment. Whether an incidentally found FLS also predicts an increased risk of arterio-arterial embolism in asymptomatic patients remains to be determined.

We compared BFl-findings with CEUS and CTA in a considerable subset of our patients. Some studies used CEUS for assessment of mural thrombus volume [12] and mural thrombus vascularization [13] in abdominal aortic aneurysms, but to our knowledge sonographic studies on WAFAS of the abdominal aorta have not been published before. We found that in the abdominal aorta, CEUS was less sensitive than CTA for the detection of WAFAS. Since such lesions in this anatomic location may have a significant clinical impact (high rate of embolic events, particularly in EVAR-patients), BFl appears as an important tool in addition to CEUS to increase the diagnostic accuracy of multimodal sonography.

By contrast, CTA was less sensitive than sonography (CEUS, BFl) for detection of WAFAS of the peripheral arteries in a limited number of our patients. Time-resolved imaging data are not available with routine CTA acquisitions. In view of this, the ability of dynamic blood flow characterization with high temporal and spatial resolution is a major strength of multimodal vascular sonography including BFl. However, the evidence regarding the frequency and the clinical impact of sonographically detected WAFAS of the peripheral arteries is sparse. Vassileva et al. reported a few cases of free-floating thrombus of the nonstenotic carotid artery detected by colour duplex sonography in ischemic stroke patients [8]. Tatheishi et al. described the “snake fang sign”, characterized by a protruding carotid plaque which was associated with free floating thrombus and ischemic stroke [14]. Our findings emphasize the advantages of a detailed sonographic examination of the peripheral arteries in cases of arterial embolism. The follow-up of WAFAS in any peripheral arterial segment can easily be performed with multimodal sonography including BFl, bearing the potential to reduce radiation exposure associated with CTA-examinations.

BFl, introduced more than 20 years ago, so far has been applied in a variety of indications, including the assessment of transplant vascularization (e.g., kidney transplants) and the dignity assessment of thyroid nodules. Studies with a focus on vascular pathologies aimed on grading of carotid stenosis [15,16] and on detection of cervical artery dissection and fibromuscular dysplasia [17,18]. A single study evaluated the diagnostic value of BFl for assessment of the lower extremity arteries and bypass grafts in patients with peripheral arterial disease [19]. In all the above mentioned vascular indications, an additional diagnostic yield of BFl has not yet been established. The addition of BFl, for example, only slightly improved the diagnostic accuracy of sonography for the diagnosis of cervical artery dissections [16]. Based on our results, we believe that the BFl-based FLS for detection of WAFAS is an important adjunct finding to established vascular imaging modalities. Whether other Doppler-independent sonographic techniques of flow detection such as superb microvascular imaging [20] offer a similar diagnostic yield for diagnosis of WAFAS remains to be determined.

Some limitations of our study warrant consideration, including retrospective data collection, the limited number of patients and the lack of complementary imaging in some cases. We are not able to provide accurate values of diagnostic accuracy of the imaging modalities analysed since we only analysed patients with an established diagnosis of WAFAS. Besides the above mentioned methodological limitations, BFl can be hampered similarly to all other ultrasound technologies by patient-related factors such as obesity, vessel calcification and intestinal gas overlay. Finally, BFl is only available with GE ultrasound machines.

In summary, we propose a clinically useful, BFl-based sonographic sign, termed FLS, for detection of WAFAS of the abdominal aorta and the peripheral arteries.

## Figures and Tables

**Figure 1 diagnostics-12-01708-f001:**
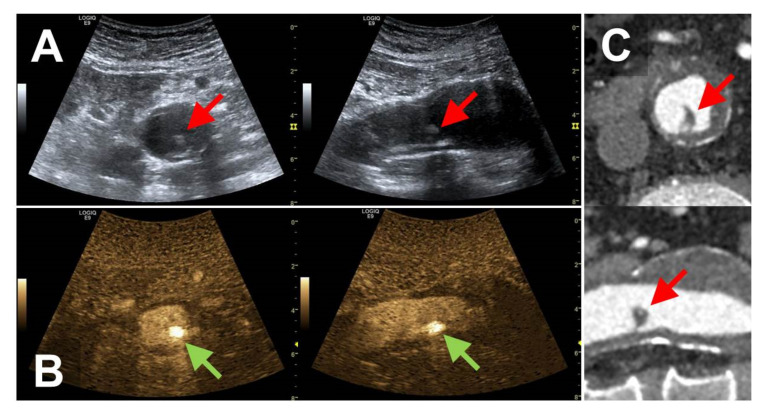
Transversal and longitudinal sections of a WAFAS (arrows) located in a small infrarenal aortic aneurysm displayed by B-mode sonography (panel **A**), BFl (FLS, panel **B**), and CTA (panel **C**). WAFAS, wall-adherent, floating arterial structure; BFl, B-Flow; FLS, flashlight sign; CTA, computed tomography angiography.

**Figure 2 diagnostics-12-01708-f002:**
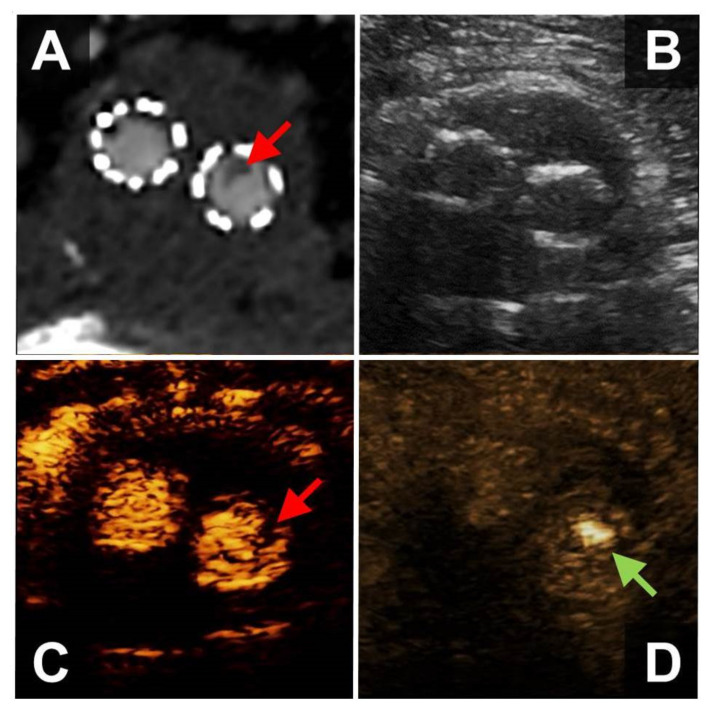
56-year-old male patient who underwent EVAR implantation for exclusion of an infrarenal abdominal aortic aneurysm. WAFAS (arrows) in the left limb of the EVAR prosthesis detectable by CTA (panel **A**), CEUS (panel **C**) and BFl (FLS, panel **D**), but not visible in B-mode sonography (panel **B**). EVAR, endovascular aortic repair; WAFAS, wall-adherent, floating arterial structure; CTA, computed tomography angiography; CEUS, contrast-enhanced ultrasound; BFl, B-Flow; FLS, flashlight sign.

**Figure 3 diagnostics-12-01708-f003:**
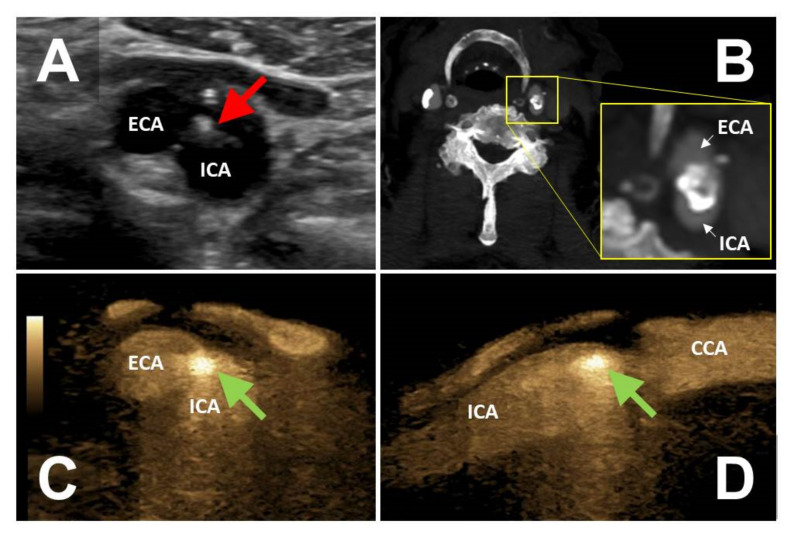
Large arteriosclerotic plaque (red arrow) resulting in ostial mid-grade stenosis of the internal carotid artery (panel **A**). CTA reveals calcification of the plaque but no evidence of WAFAS (panel **B**). WAFAS was discovered by BFl (FLS, green arrows) in the transversal (panel **C**) and longitudinal (panel **D**) plane. CTA, computed tomography angiography; WAFAS, wall-adherent, floating arterial structure; BFl, B-Flow; FLS, flashlight sign; ECA, external carotid artery; ICA, internal carotid artery; CCA, common carotid artery.

**Table 1 diagnostics-12-01708-t001:** Clinical characteristics and treatment of symptomatic patients.

Patient Characteristics	FLS Location	Clinical Event	Treatment
Male, 58 years	Native abdominal aorta	Left sided popliteal artery occlusion	Bypass surgery
Male, 66 years	EVAR, left limb	Left sided popliteal artery occlusion	Catheter thrombectomy
Male, 67 years	EVAR, main body and right limb	Metachronous bilateral popliteal artery occlusions	Catheter thrombectomy
Male, 56 years	EVAR, left limb	Left sided popliteal artery occlusion	Catheter thrombectomy
Male, 74 years	EVAR, both limbs	Metachronous, bilateral popliteal artery occlusions	Catheter thrombectomy; subsequent re-lining of the EVAR prosthesis
Female, 68 years	Brachiocephalic trunk	Branch retinal artery occlusion	Dual antiplatelet therapy
Male, 80 years	Left axillary artery	Digital artery occlusion with critical finger ischemia	Percutaneous transluminal angioplasty of the underlying stenosis

FLS, flashlight sign; EVAR, endovascular aortic repair.

**Table 2 diagnostics-12-01708-t002:** Characteristics of the overall cohort and certain subgroups.

		Overall Cohort *n* = 28	Group A (Native Abdominal Aorta) *n* = 8	Group B (Endovascular Aortic Repair) *n* = 10	Group C (Peripheral Arteries) *n* = 10
**Indication for ultrasound examination**					
Thrombembolic event (*n*; %)		7 (25)	1 (12.5)	4 (40)	2 (20)
Incidental finding (*n*; %)		7 (25)	5 (62.5)	/	2 (20)
Routine follow up after vascular procedure (*n*; %)		14 (50)	2 (25)	6 (60)	6 (60)
**Complementary imaging**					
**CEUS**	Total (*n*; %)	13 (46.4)	1 (12.5)	9 (90)	3 (30)
	Positive (*n*; %)	8 (28.6)	/	6 (60)	2 (20)
	Negative (*n*; %)	5 (17.9)	1 (12.5)	3 (30)	1 (10)
	PPV of the FLS in comparison to CEUS (%)	61.5	0	66.7	66.7
**CTA**	Total (*n*; %)	17 (60.7)	6 (75)	8 (80)	3 (30)
	Positive (*n*; %)	13 (46.4)	6 (75)	7 (70)	/
	Negative (*n*; %)	4 (14.3)	/	1 (10)	3 (30)
	PPV of the FLS in comparison to CTA (%)	76.5	100	87.5	0

CEUS, contrast enhanced ultrasound; CTA, computed tomography angiography; PPV, positive predictive value.

## Data Availability

The data presented in this study are available on request from the corresponding author.

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
