# Peer review of "The Flashlight-Sign: A Novel B-Flow Based Ultrasound Finding for Detection of Intraluminal, Wall-Adherent, Floating Structures of the Abdominal Aorta and Peripheral Arteries"

_diagnostics, 2022, doi:10.3390/diagnostics12071708_

Round 1

Reviewer 1 Report

In a retrospective study with an indeed limited number of patients (n=28 in a six-year period) th authors describe a novel sonographic sign (flashlight sign) in B-Flow imaging in different vascular regions, which shows floating intravascular strucutres. This new observation , to my opinion is very helpful for vascular ultrasound and the technique is superior to CTA and contrast enhanced ultrasound. In summary, this solid presentation is of clinical importance and should be open to the scientific public.

Author Response

No revisions requested.

Reviewer 2 Report

In this manuscript, the Authors describe diagnosis of mobile structures adherent to the vascular wall by a non-Doppler ultrasound functionality called B-flow. They mentioned dense visualization of the movement by the floating structure. 

Generally, it is interesting. Unfortunately, the number of included patients is quite low and no detailed statistical analysis (comparison with other methods) would be possible.  Therefore, it is rather a case series and according to my opinion, it should be presented in this way. Or, to perform an in-vitro study/model.

For the users of GE devices, BFI is the B-flow combined with color Doppler, therefore, the abbreviation "BFI" is not ideal. 

In Figure 3 legend, panel D is not mentioned.

The references are rather sparse and old. 

Author Response

In this manuscript, the authors describe diagnosis of mobile structures adherent to the vascular wall by a non-Doppler ultrasound functionality called B-flow. They mentioned dense visualization of the movement by the floating structure. 

R2.1 Generally, it is interesting. Unfortunately, the number of included patients is quite low and no detailed statistical analysis (comparison with other methods) would be possible.  Therefore, it is rather a case series and according to my opinion, it should be presented in this way. Or, to perform an in-vitro study/model.

The limited sample size is mentioned as important limitation in the manuscript. The idea of an in vitro study model is intriguing and something worthwile to work on in the future, but currently we are not able to provide such data.

R2.2. For the users of GE devices, BFI is the B-flow combined with color Doppler, therefore, the abbreviation "BFI" is not ideal. 

We agree and changed the abbreviation “BFI” to “BFl” throughout the manuscript.

R2.3. In Figure 3 legend, panel D is not mentioned.

Thank you for the careful review of the manuscript. We added the legend to panel D.

R2.4. The references are rather sparse and old. 

We updated our literature search in Pubmed but except for single case reports (in their majority reporting floating thrombus of the ascending aorta and the carotid arteries) we found no recent citations with relevance for our paper. However, we added another reference meaningful for the discussion of our findings (Ref. 14 in the revised manuscript).

Reviewer 3 Report

In this manuscript Lottspeich et al. want to "to determine the potential diagnostic yield of B-Flow Imaging (BFI)-based signl in the detection of wall-adherent, floating arterial structures (WAFAS) in various anatomical locations and to compare BFI with contrast enhanced ultrasound (CEUS) and computed tomography angiography (CTA). Interestingly, they suggest BFI-based sonographic sign, termed FLS, for detection of WAFAS of the abdominal aorta and the peripheral arteries.

The following points must be considered:

1) The sample size seems too low to reach final results. A sample size calculation must be provided.

2) No statistical comparison was made between the different imaging methods. This makes it difficult to quantify any differences.

3) The retrospective nature of the study is a pivotal limitation.

4) Clinical features, therapy, timing of radiological investigations must be provided to the reader. I suggest that you include a table in this regard.

5) Correct typos in the text (e.g. CDUS)

Author Response

In this manuscript Lottspeich et al. want to "to determine the potential diagnostic yield of B-Flow Imaging (BFI)-based signl in the detection of wall-adherent, floating arterial structures (WAFAS) in various anatomical locations and to compare BFI with contrast enhanced ultrasound (CEUS) and computed tomography angiography (CTA). Interestingly, they suggest BFI-based sonographic sign, termed FLS, for detection of WAFAS of the abdominal aorta and the peripheral arteries.

The following points must be considered:

R3.1. The sample size seems too low to reach final results. A sample size calculation must be provided.

According to our understanding, a sample size calculation aims on providing adequate statistical power for significance tests of prospective studies. This is a retrospective observational study, and testing for statistical significance was not performed. Given these considerations, we do not provide a sample size calculation.

R3.2. No statistical comparison was made between the different imaging methods. This makes it difficult to quantify any differences.

Please see response to 3.1. Given the limited number of cases, statistical significance tests do not seem to be useful.

R3.3. The retrospective nature of the study is a pivotal limitation.

That the study was performed retrospectively is mentioned as limitation in the manuscript.

R3.4. Clinical features, therapy, timing of radiological investigations must be provided to the reader. I suggest that you include a table in this regard

With regard to the time intervals between Bfl, CEUS, and CTA, additional information is provided in the revised manuscript. Information on clinical features and treatment of symptomatic patients is provided in an additional table.

R3.5. Correct typos in the text (e.g. CDUS)

Thank you for the careful review of the manuscript. The typos was corrected.

Round 2

Reviewer 2 Report

I am satisfied by the Authors´responses